# The Influence of Socioeconomic Factors on the Body Characteristics, Proportion, and Health Behavior of Children Aged 6–12 Years

**DOI:** 10.3390/ijerph20043303

**Published:** 2023-02-13

**Authors:** Joanna Nieczuja-Dwojacka, Beata Borowska, Alicja Budnik, Justyna Marchewka-Długońska, Izabela Tabak, Katarzyna Popielarz

**Affiliations:** 1Institute of Biological Sciences, Faculty of Biology and Environmental Sciences, Cardinal Stefan Wyszynski University in Warsaw, 01-938 Warsaw, Poland; 2Department of Anthropology, Faculty of Biology and Environmental Protection, University of Lodz, 90-136 Lodz, Poland

**Keywords:** development, health, body proportions, children, socioeconomic status

## Abstract

Background: The research aimed to determine how socioeconomic factors influence the body structure and health behaviors of children in a suburban commune. Methods: Data from 376 children aged 6.78 to 11.82 years from Jabłonna, Poland, were analyzed. A questionnaire was used to gather information regarding the socioeconomic status and dietary habits of these children, and physical measurements such as height, weight, pelvic width, shoulder width, chest, waist, hip, and arm circumferences, and three skinfolds were taken. Hip index, pelvi-acromial index, Marty’s index, BMI (body mass index), WHR (waist–hip ratio), and the sum of three skinfolds were calculated. One-way analysis of variance, Student’s *t*-test, and *X*^2^ test with *p <* 0.05 were used. Results: The size of the family and the level of education and occupation of the fathers had a significant impact on the body proportions of the children. Children from larger centers with more educated parents were seen to have healthier eating habits and higher levels of physical activity, and their parents were less likely to smoke cigarettes. Conclusions: It was concluded that the development environment of the parents, such as their level of education and profession, play a more important role than the size of birthplace.

## 1. Introduction

Human development is influenced by a variety of factors, such as genetics, maternal influences, and the environment. The environmental factors can be categorized as natural and socioeconomic, and the latter includes, among others, the place of birth, the place of residence, education and profession of parents, and the size of the family [1,2].

Socioeconomic status (SES) is considered to be an important factor in the development of children and adolescents and is a modifier of biological characteristics such as height and weight [3,4]. It has also been reported that environmental factors may have a greater effect on lower limb length than genetic factors [5]. 

It has been observed that children living in cities are characterized by a greater body height than those living in rural areas. It is also observed that children whose parents have higher education are characterized by a higher body height than children whose parents have an average or lower level of education. A similar pattern is observed with regard to the parents’ profession—the higher the salary of the specialized job, the more often the children are characterized by higher body height. The height of the body largely depends on the length of the lower limbs, and the length of the lower limbs has been shown to have a positive correlation with the family’s monthly income [6,7].

In the case of body weight, the tests are inconclusive. The current overweight and obesity epidemic is associated with the consumption of excessive calories and mediocre levels of physical activity [8]. Children whose parents have a higher education level, hold better professions, and live in cities should theoretically have a lower weight, and their BMI (Body Mass Index) should be normal. However, this view is not supported by many studies [9].

Since BMI is not free from errors related to body composition, it is important to consider other parameters of the massiveness of the body structure, such as the circumference of the arm or skinfolds. In addition, BMI may vary depending on the population and its characteristic body proportions [10,11,12].

Several studies have reported the influence of SES on the body structure and proportions of children and adolescents. Comparative studies of children living in villages and towns are often cited. It has been observed that parents’ education level has an influence on the shoulder width and circumference of children, and the children of fathers with a higher level of education showed an endomorphic structure [6].

SES also seems to play a role in promoting a healthy lifestyle and health behaviors. It is noted that higher SES is associated with better health status, a better, as well as more varied, diet, and more frequent involvement in sports [13,14,15,16]. However, results concerning the influence of socioeconomic factors on the consumption of alcohol and stimulants are not unequivocal [17,18].

All the above-mentioned factors do not directly influence the body height or health behavior of children, but are conducive to the creation of an appropriate development environment, for example, through parental awareness, selection of good housing conditions, access to health care, and greater attention to proper diet and lifestyle, including the correct level of physical activity [19].

The present study focused on an interesting location—the rural commune of Jabłonna, where a large number of people work in Warsaw (capital city) and are mostly characterized by a relatively high level of education. Therefore, the aim of the study is to identify the SES-related factors that affect the structure and proportions of the body, as well as the pro-health behavior of children from the Jabłonna commune near Warsaw, and to determine whether the effects of the rural or urban environment are more pronounced in towns near large urban centers.

Such studies are extremely valuable because the biological condition of children and adolescents is a determinant of their health, and development modifiers can contribute to the wellbeing of the population.

## 2. Materials and Methods

The study was carried out on a group of 376 children, including 162 boys and 214 girls, with a mean age of 9 (6.78–11.82) and 9.30 (6.80–11.79) years, respectively. Three primary schools from Jabłonna commune were randomly selected from the studied voivodship in 2018. Jabłonna is rural commune in Mazovia Voivodeship, with a population of 22,289 inhabitants in 2021, population density of 344/km^2^. During the period of 2011–2021, the annual population change was 3.1% [20].

The study was approved by the Committee for the Bioethics of Scientific Research at the University of Lodz. Consent from parents and children was also obtained. 

Data inclusion criteria were signed approval and age from 6 to 12 years. The level of sexual maturation was determined through interview, and only children with no symptoms of puberty were selected. Exclusion criteria included lack of SES data or anthropometry measurements.

Parents participating in the study completed a questionnaire with closed questions regarding the SES: place of birth (village/town/city), the number of family members living together (2–3, 4–5, or 6 or more people), level of education (primary and vocational, secondary, university), and occupation (unqualified, technical, specialist). The above questions were used to determine the SES of families participating in the study [21]. In addition, parents were also surveyed about their smoking habit (yes/no), children’s involvement in additional sports (yes/no), and nutritional conditions (bad/medium/very good). Additionally, children were asked about the consumption of a given product (fruit, vegetables, sweets) during the week. In order to standardize the responses, the possible options for consumption of a given product were given as follows: rarely (0–2 times a week), on average (3–4 times a week), or often (5 or more times a week). The above questions were used to determine the lifestyle and health behavior of families. All measurements were taken by an experienced anthropologist, according to the generally accepted technique [22]. The measuring instruments were standardized and regularly calibrated before measurements, and their accuracy was checked during the measurement. Technical errors of measurement were determined by a sample study performed on 10% of children. Each of these measurements were taken in duplicate. Only published standards were used for the study [23]. The body height (B-v) was measured with an anthropometer (GMP) to the last 0.1 cm with the children in a standing position. Body weight was measured using a medical scale (WPT 60/150 OW, RADWAG) with an accuracy of 0.1 kg. Pelvic width and shoulder width were measured using a large spreading caliper (GMP) to the last 0.1 cm. Chest, waist, hip, and arm circumferences were measured using a tape (SECA) with a precision of 1 mm. Using a skinfold caliper (Harpenden), subcutaneous skinfold on arm, subscapular skinfold, and abdominal skinfold were measured, and the sum of these three skinfolds was calculated [22,24]. All the anthropometric measurements were taken in a private room. Based on the obtained measurements, the following body proportions indexes were calculated: hip index [(bi-iliac width/height) × 100], pelvi-acromial index [bi-iliac width/bi-acromial width) × 100], Marty’s index [(chest circumference/height) × 100], BMI, and WHR [25,26]. Age was calculated as the difference between the date of measurement and the date of birth.

The obtained measurements were transferred to the *Z* scale to analyze the entire studied population, regardless of age, with division into sex [27]. One-way analysis of variance and Student’s *t*-test (*p* < 0.05) were used for statistical analysis. Normal distribution was checked and confirmed. To verify whether socioeconomic factors had an influence on the lifestyle and health behavior, *X*^2^ test was used (*p* < 0.05) for the entire studied population, regardless of sex.

## 3. Results

### 3.1. Statistical Characteristics of the Surveyed Population

The vast majority of the surveyed children were born in a city or town, while 7% were born in rural areas. Approximately 73% of the children were brought up in 4–5 person families, 21% in 2–3 person families, and another 6% belonged to families with more than 6 people. Among the surveyed parents, 75% had higher education, while 25% had secondary education. Due to the very small size of the groups with lower levels of education, these data were not considered in the statistical analysis. Over 61% and 18% of parents held “specialist” and “qualified and technical” positions, respectively. More than 21% of all fathers were characterized by “unskilled” occupations, while mothers had occupations as “specialist” and “qualified and technical”. The smallest group were the unemployed (3%), and their data were not taken into account in the statistical analysis.

Among the surveyed fathers, over 22% declared a smoking habit, while among mothers, 16% admitted that they smoked. Among the surveyed children, 58% participated in additional physical activities. Most parents described their nutritional conditions as good (81%) and almost 19% as average, and none declared bad nutritional conditions. Most of the children (over 67%) consumed sweets frequently (more than 5 times a week), over 27% of the respondents admitted eating sweets 3–4 times a week, and about 5% of children reported that they rarely ate sweets.

### 3.2. Results of the Statistical Analysis

There was no significant difference between children living in cities and those born in towns and villages, in terms of body proportion.

The size of the family showed a correlation with the sum of three skinfolds in the case of girls. Girls growing up in small families had a significantly higher body weight compared to those living in larger families. Schoolgirls from families of 2–3 people showed higher values of Marty’s index compared to those from families of 4–5 people. Similarly, the values of BMI, WHR, arm circumference, and the sum of three skinfolds were higher among girls belonging to families of 2–3 people than those from families of 4–5 people or more. However, such a correlation was not observed in boys.

Father’s level of education had no impact on the body proportions of the examined boys, while in the case of girls, there were differences observed in the hip index, pelvi-acromial index, Marty’s index, and body weight. Girls whose fathers had a lower level of education were characterized by higher values of the above indices, as well as higher body mass values. Moreover, the lower the level of the father’s education, the greater the values of BMI, WHR, arm circumference, and the sum of three skinfolds among girls. On the other hand, the mother’s level of education did not significantly affect the studied variables in both sexes. 

Father’s occupation had a significant influence on the value of the arm circumference in boys. Boys whose fathers worked as technicians showed the largest arm circumference, while the children of specialists had the smallest arm circumference. Mother’s occupation did not significantly influence the body structure and proportions in boys. The results of the statistical analysis are presented in Table 1 and Table 2.

The number of family members had a significant influence on smoking habit among both mothers (*X*^2^ = 10.375; *p* = 0.005) and fathers (*X*^2^ = 6.557; *p* = 0.037). The other studied variables did not significantly affect the behavior and lifestyle of the respondents.

Fathers’ level of education significantly influenced smoking among caregivers (*X*^2^ = 16.532; *p* = 0.000) and the participation of children in additional sports (*X*^2^ = 11.25; *p* = 0.003). Similarly, mother’s level of education significantly influenced the partner’s smoking habit (*X*^2^ = 6.328; *p* = 0.011) and the involvement of children in additional sports (*X*^2^ = 8.646; *p* = 0.003). The higher the mother’s or father’s education level, the less frequently the fathers smoked. On the other hand, the higher the parents’ level of education, the more often the children practiced additional sports.

The profession of mothers did not significantly affect the studied variables, while father’s profession significantly influenced smoking by caregivers (*X*^2^ = 10.408; *p* = 0.005), and specialists smoked less frequently. Children of specialists were significantly more likely to play sports (*X*^2^ = 9.058; *p* = 0.010), and their nutritional conditions were better (*X*^2^ = 10.443; *p* = 0.005).

The *X*^2^ test showed that the habit of fruit consumption differed significantly among children from centers of various sizes (*X*^2^ = 10.58; *p* = 0.032).

Other factors did not significantly affect the examined dietary and health habits.

## 4. Discussion

A comparison of the values of indicators and measurements in boys and girls revealed that the father’s occupation had an influence on the values in boys, while the size of the family and the father’s level of education had a significant effect on the variables in girls. Interestingly, neither the place of birth nor the mother’s level of education or her profession had any significant impact on any of the analyzed parameters.

Rural areas in Poland comprise over 90% of the country’s land area, and are inhabited by almost 40% of the population. However, the number of people employed in the agricultural sector continues to decline. There are two types of migration—people moving from the cities to the countryside and vice versa [28]. It is assumed that the place of origin may have an effect on the physique of children and adolescents, although more and more researchers note that this factor is no longer as important and the differences between urban and rural areas have been leveled, which is especially noticeable in developed countries [29]. In the case of developing countries, such differences can still be observed, due to, inter alia, poorer access to better paid jobs, poorer living conditions, and poorer access to food among people living in the countryside. In addition, earlier studies show that in developing countries, when living conditions improve, for example in the case of economic migration, the body proportions of children transform. An example is the research on Maya families from Guatemala who migrated to the United States. People who grew up in the USA were taller and had longer legs than their parents, which could be a result of better access to food [5]. In Poland, slightly later than in Western European countries, there were major social and economic changes that resulted in changes in lifestyle, access to modern medicine, and an increased awareness of the development of children and adolescents [30,31]. In Poland, from 1989 to 2002, migrations from the countryside to cities were recorded, but now the opposite is noted. Mostly, wealthy people migrate looking for more ecologically diverse areas, giving them the possibility of recreation and rest outside the city. Many of these people work in a big city. In the last 30 years, the population in suburban areas has been increasing. In Poland, there was not a statistically significant correlation observed between the number of overweight and obese children and the urbanization level [32,33].

The size of the center from which the examined children came did not significantly differentiate their lifestyle and health behaviors. One difference was noted in their frequency of eating fruits. This could be due to the availability of seasonal fruits in rural areas. Reports in the literature indicate that limited access to fruit may be influenced by factors such as income, degree of urbanization, level of education, and land use [34]. Furthermore, lack of knowledge and socioeconomic inequalities may be influencing factors in the consumption of fewer fruits and vegetables than recommended [35].

The second parameter considered was the size of the family. In Poland, a higher fertility rate was recorded for families living in villages, although recent reports show that rural areas in Poland did not attain a rate of 2.1 children per woman of reproductive age. This is explained by the demographic transition that began in Poland in the 1990s [36]. Among the examined children, it was found that in smaller families, girls had higher values of Marty’s Index, BMI, WHR, arm circumference, sum of three skinfolds, and body weight. This factor did not affect the boys’ body proportions. It is suggested that limited family resources may be a factor, as in smaller families, there may be more resources available for each family member, mainly when it comes to access to food. It should also be noted that in Poland fast food and sweets are relatively expensive and can be purchased only by families with greater financial resources and fewer children. This fact may explain why this factor affects the body structure of children, although in the case of the results concerning the influence of family size on lifestyle and health behavior, most significant differences were not observed [37]. However, this factor significantly influenced the smoking habits of the fathers and mothers of the examined children, with differences observed in the case of smaller and medium-sized families. Reports in the literature conclude that parents who smoke more frequently come from families with lower SES. This is due to the lack of awareness of the effects of tobacco smoke on the developing organism of a child and the health consequences, such as increased incidence of lung cancer [38,39]. 

Parental education is a frequently cited factor in SES. In the case of children from Jabłonna, the father’s education level significantly influenced the body proportions of girls, while the mother’s level of education did not influence the girl’s body proportions. In the survey of fathers who had a lower level of education, the girls had higher values of hip index, pelvi-acromial index, Marty’s index, body weight, skinfolds, arm circumference, BMI, and WHR. 

The results of the *X*^2^ analysis indicated that children of parents with higher levels of education were more likely to engage in sports than their peers whose parents had lower levels of education. This may be attributed to the fact that parents with higher levels of education are more knowledgeable about the importance of healthy eating habits and physical activity, as well as the health status of their children, and may have the resources to adequately support their children’s involvement in sports. The current research does not confirm a direct relationship between the education level of mothers and their children’s lifestyle. However, other studies have noted that a mother’s education level may have an influence on her child’s lifestyle, particularly in overweight children [40]. Moreover, the results of many studies conclude that lower maternal education is associated with less healthy food choices in children. More income can also contribute to changes in lifestyle—children from families with better SES are more likely to have access to, for example, extra-curricular sports than those from poorer families [41,42]. The results of other studies confirm that higher values of BMI, WHR, and endomorphic type are associated with lower levels of parental education [6,43,44]. The results of the current study also suggest that higher education levels of mothers and fathers are linked to less smoking among caregivers. This is in line with other research results, which can be attributed to a higher level of parental awareness gained from higher education. Interestingly, the level of education of mothers was not related to the children’s opinions about smoking, although not all research results are unequivocal. It is therefore important to educate and make children aware of the consequences of smoking [45].

There is a high correlation between the level of education and the profession pursued by a parent, with a higher level of education often associated with better income. Studies show that the arm circumference of boys is directly impacted by their father’s occupation, with those whose fathers had technical professions having the highest circumference and those with fathers employed as specialists having the lowest circumference. No such dependencies were found in girls. Reports by other authors suggest that overweight or obesity in children may be related to parental occupation, especially of fathers [46]. In the case of the analysis of the impact of socioeconomic factors on the lifestyle and health behavior of children, it was noted that the children of fathers employed as specialists engaged more in different sports and had adequate nutrition. Moreover, fathers employed as specialists smoked cigarettes less frequently. The current results can also be seen in reports by other authors [47]. SES factors play a crucial role in the prevalence of poor nutritional conditions in children [48]. A large body of evidence shows a relationship between low maternal education levels, poverty, and malnutrition [49,50].

In large cities and relatively wealthy regions, rural areas may look different from typically rural areas located in the northern or eastern parts of a country. This type of research is important, as it not only highlights the effects of certain factors on the development of children and adolescents, but also illuminates the inequalities caused by SES stratification.

## 5. Conclusions

The presented results and literature data indicate that some of the SES factors may have an influence on the body structure and proportions, lifestyle, and health behavior of children, including their sports involvement and feeding. 

The study showed a high correlation between the level of education and the profession of parents, with a higher level of education often associated with better income. In the case of girls from Jabłonna, father’s education level significantly influenced the body structure. Previous studies have shown that in developing countries, the body proportions of children improved with better living conditions, such as economic migration. Currently, the size of the center does not play an important role in the development environment, while parents’ level of education and their profession are more important. 

Thus, it is necessary not only to conduct further research on these problems but also to implement preventive initiatives related to nutrition and sports participation of children.

The present study has certain limitations, such as a small sample size and lack of data about family income. In Poland, family income is considered as personal information, and, according to our previous experience, it is often the reason cited for nonparticipation in research.

The aim of the researchers is to perform further studies comparing rural and urban groups in order to verify the impact of sociodemographic factors on the lifestyle and behavior of children.

## Figures and Tables

**Table 1 ijerph-20-03303-t001:** Results of one-way analysis of variance showing the relationship between selected socioeconomic factors and indices of body composition and body proportions in boys and girls (* *p* < 0.05).

	Boys					Girls				
	SS	df	MS	*F*	*p*	SS	df	MS	*F*	*p*
	**Place of birth**									
Hip index	0.1469	2	0.073	0.072	0.930	0.136	2	0.068	0.066	0.935
Pelvi-acromial index	2.3558	2	1.177	1.174	0.310	0.513	2	0.256	0.252	0.776
Marty’s index	0.7858	2	0.392	0.388	0.678	0.454	2	0.227	0.218	0.803
BMI	2.383	2	1.191	1.189	0.306	2.593	2	1.296	1.293	0.277
WHR	0.500	2	0.250	0.223	0.799	0.619	2	0.309	0.305	0.737
Arm circumference	2.761	2	1.380	1.375	0.254	0.451	2	0.225	0.222	0.801
Sum of three skinfolds	0.378	2	0.189	0.186	0.829	3.469	2	1.734	1.635	0.198
Body height	5.30	2	2.650	0.045	0.955	0.898	2	0.449	0.442	0.643
Body mass	1010.6	2	505.298	0.133	0.875	1.840	2	0.920	0.913	0.403
	**Family size**									
Hip index	1.749	2	0.874	0.870	0.420	2.152	2	1.076	1.067	0.346
Pelvi-acromial index	0.602	2	0.300	0.297	0.742	2.505	2	1.252	1.248	0.289
Marty’s index	1.749	2	0.874	0.868	0.421	6.276	2	3.13	3.125	0.046 *
BMI	2.950	2	1.475	1.477	0.230	10.505	2	5.252	5.512	0.004 *
WHR	4.947	2	2.473	2.253	0.107	8.081	2	4.040	3.917	0.021 *
Arm circumference	1.862	2	0.931	0.923	0.398	8.832	2	4.416	4.573	0.011 *
Sum of three skinfolds	4.087	2	2.043	2.054	0.130	8.503	2	4.251	4.403	0.013 *
Body height	1.438	2	0.719	0.713	0.490	1.311	2	0.655	0.649	0.523
Body mass	2.384	2	1.192	1.188	0.306	7.480	2	3.740	3.840	0.023 *
	**Father’s occupation**									
Hip index	2.831	2	1.415	1.416	0.244	1.067	2	0.533	0.526	0.591
Pelvi-acromial index	2.158	2	1.079	1.075	0.343	4.664	2	2.332	2.355	0.098
Marty’s index	1.886	2	0.943	0.937	0.393	1.786	2	0.893	0.865	0.422
BMI	5.011	2	2.505	2.533	0.081	4.172	2	2.086	2.101	0.125
WHR	2.019	2	1.009	0.908	0.404	3.080	2	1.540	1.448	0.237
Arm circumference	6.672	2	3.336	3.386	0.035 *	0.693	2	0.346	0.341	0.711
Sum of three skinfolds	1.456	2	1.456	1.453	0.229	1.250	2	0.6251	0.618	0.540
Body height	4.687	2	2.343	2.362	0.096	0.809	2	0.404	0.399	0.671
Body mass	2.087	2	1.043	1.039	0.355	2.519	2	1.259	1.256	0.287

SS—sum of Squares; df—degrees of freedom; Ms—mean square.

**Table 2 ijerph-20-03303-t002:** Differences in the body proportions of boys and girls depending on their parents’ education and mother’s profession; Student’s *t*-test results (* *p* < 0.05).

	Boys					Girls				
	Mean 1	Mean 2	*t*	df	*p*	Mean 1	Mean 2	*t*	df	*p*
**Father’s education level**										
Hip index	−0.079	0.171	−1.707	212	0.089	0.227	−0.162	2.476	160	0.014 *
Pelvi-acromial index	−0.087	0.193	−1.919	212	0.056	0.244	−0.176	2.688	160	0.007 *
Marty’s index	−0.005	0.045	−0.348	212	0.728	0.198	−0.154	2.207	160	0.028 *
BMI	−0.033	0.075	−0.735	212	0.462	0.358	−0.259	4.050	160	0.000 *
WHR	−0.004	0.048	−0.335	212	0.737	0.284	−0.148	2.679	160	0.008 *
Arm circumference	−0.073	0.163	−1.606	212	0.109	0.247	−0.178	2.720	160	0.007 *
Sum of three skinfolds	−0.079	0.174	−1.725	212	0.085	0.257	−0.186	2.837	160	0.005 *
Body height	0.025	−0.055	0.543	212	0.587	−0.007	0.005	−0.085	160	0.932
Body mass	0.026	−0.057	0.568	212	0.570	0.257	−0.186	2.841	160	0.005 *
**Mother’s education level**										
Hip index	−0.053	0.238	−1.656	212	0.099	0.096	−0.018	0.551	160	0.582
Pelvi-acromial index	−0.022	0.102	−0.706	212	0.480	0.088	−0.018	0.511	160	0.610
Marty’s index	−0.029	0.191	−1.249	212	0.212	0.271	−0.064	1.597	160	0.112
BMI	−0.030	0.139	−0.957	212	0.339	0.290	−0.061	1.697	160	0.091
WHR	0.043	−0.128	0.922	212	0.357	0.322	−0.027	1.635	160	0.103
Arm circumference	−0.020	0.095	−0.650	212	0.516	0.079	−0.016	0.459	160	0.646
Sum of three skinfolds	−0.034	0.153	−1.060	212	0.289	0.010	−0.001	0.059	160	0.952
Body height	0.020	−0.092	0.637	212	0.524	0.093	−0.019	0.542	160	0.588
Body mass	0.023	−0.104	0.719	212	0.472	0.256	−0.053	1.491	160	0.137
**Mother’s occupation**										
Hip index	−0.033	0.047	−0.580	212	0.562	0.004	−0.000	0.032	160	0.973
Pelvi-acromial index	0.020	−0.029	0.353	212	0.723	0.049	−0.025	0.453	160	0.650
Marty’s index	−0.010	0.041	−0.374	212	0.708	0.081	−0.052	0.800	160	0.424
BMI	−0.049	0.075	−0.895	212	0.371	0.101	−0.053	0.933	160	0.351
WHR	−0.062	0.122	−1.260	212	0.208	0.191	−0.050	1.424	160	0.156
Arm circumference	−0.083	0.126	−1.504	212	0.133	0.134	−0.070	1.241	160	0.216
Sum of three skinfolds	−0.067	0.100	−1.205	212	0.229	0.030	−0.015	0.275	160	0.783
Body height	0.015	−0.023	0.281	212	0.778	0.031	−0.016	0.290	160	0.771
Body mass	0.011	−0.016	0.195	212	0.844	0.126	−0.067	1.172	160	0.242

df—degrees of freedom.

## Data Availability

The data presented in this study are available on request from the corresponding author. The data are not publicly available due to privacy.

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
