# Peer review of "The Influence of Socioeconomic Factors on the Body Characteristics, Proportion, and Health Behavior of Children Aged 6–12 Years"

_ijerph, 2023, doi:10.3390/ijerph20043303_

Round 1
Reviewer 1 Report
The Authors decided to undertake a fundamental problem of the association between socioeconomic factors and body size and proportions. It is always a current issue, as the environment in which human populations develop constantly changes. However, in the Reviewer’s opinion, the manuscript needs significant corrections before being considered for publication.
Detailed comments and suggestions are provided below.
· Abstract:
o what do the Authors understand as the “length”? If it was body height, it should be called so, as the length and height of the body are two entirely different measurements;
o ll.24-25 – the Authors refer to the “centres of various sizes”; it is very unclear what it means, considering previous information, that the study group came from Jabłonna (a single location);
o the journal’s guidelines for the abstract should include the following: 1) Background: Place the question addressed in a broad context and highlight the purpose of the study; 2) Methods: Describe briefly the main methods or treatments applied. Include any relevant preregistration numbers and species and strains of any animals used. 3) Results: Summarize the article's main findings; and 4) Conclusion: Indicate the main conclusions or interpretations; thus, the abstract is missing the “Background” section; additionally, the 200 words limit should be taken into consideration while correcting this section of the text;
o in the Reviewer’s opinion, the phrase “body build” used in line 24 should be replaced by body proportions, measurements and/ or characteristics;
o is the conclusion about SES not influencing body size and proportions of the examined children justified, considering the statistically significant differences observed by the Authors?
· Introduction:
o BMI and SES abbreviations should be explained upon the first use in the text;
o ll.51-55 – how does the information provided in the paragraph coincide with the fact that after countries go through the demographic transition, there is also a shift in the socioeconomic strata, which experience the heaviest burden of overweight/ obesity?
o ll. 81-82 – the Authors mention large urban and town areas; how does it correspond with the previously mentioned Jabłonna?
o the Authors should provide more information regarding the research site (e.g. location, characteristics of the town, etc.);
· Material and methods:
o the socioeconomic parameters collected in the study are, unfortunately, very poorly explained; for instance, it is not clear what is meant by:
§ place of birth – presumably, it refers to the children’s birthplace; how does this correspond with the area and socio-economic environment in which the child is actually being raised? moreover, how it corresponds to the fact (mentioned earlier in the text) that in Jabłonna, many residents spend a significant amount of time in Warszawa?
§ town, village or city (were there particular urbanisation indicators provided to choose form or number of residents?);
§ parents’ occupation, what exactly is a specialist job? how does it differ from a technical one?
§ nutritional status – how exactly was it assessed, what was taken into consideration, especially since there were also questions regarding the frequency at which the children consumed particular foods;
o l.111 – “generally accepted method” – different “standard” techniques still have methodological differences; thus, it should be specified which one was used (for instance, Martin’s, IBP etc.);
o l. 116 - as mentioned before, the length and height of the body are two separate measurements; thus, the sentence should be corrected appropriately;
o l.120 – what type of skinfold calliper was used? additionally, information regarding the manufacturers of all used tools should be provided;
o l.121 – which “arm” skinfold was measured?
o l.131-133 –what test was used to assess the normality of the distribution;
· Results:
o the decimal places in the tables should be separated by dots, not commas;
o l. 164-165 –the Authors mention the body composition, but there is no mention of measuring that (besides the sum of three skinfolds) in the text; also, the sentence concerning the 2-3-person families is very unclear;
o the entire description of the results of the Chi2 analysis should be rewritten to be more clear and understandable for the reader, as they are, unfortunately, quite chaotic and unclear;
· Discussion:
o once again, is the conclusion about SES not influencing body size and proportions of the examined children justified, considering the statistically significant differences observed by the Authors?
o the mandatory “Conclusions” section is missing;
· References: the list should be made more consistent, as there are differences in how the records are formatted (e.g. abbreviated vs the extended version of the name of the journal);
· General remarks:
o names of the indicators and tools used in the study should be checked and corrected accordingly (e.g. it should be pelvi-acromial index, not acromio-iliac, Marty’s, not Marty, large spreading calliper, not large calliper etc.);
o the entire article should be carefully proofread; grammatical mistakes should be corrected
o the Authors should also use language appropriate for scientific publication (e.g. taller body is not a proper scientific term).
Author Response
Odpowiedź na recenzenta 1 Komentarze

Reviewer 2 Report
The introduction provides adequate background, but there is a lack of more contemporary literature on the problem under investigation, and I propose to refine the introduction and discussion with a few more relevant references from the last five years.
The research design is appropriate and the methods chosen are appropriate in relation to the goal to be achieved.
The results are very clearly presented and comprehensibly described.
The conclusions are clearly stated and are based on the results obtained.
I suggest that in the conclusion, the scientific contribution of the results and the proposal for the continuation of the research should be stated more clearly.
Author Response
Response to Reviewer 2 Comments in pdf.

Reviewer 3 Report
Comments and suggestions are attached.

Author Response
Response to Reviewer 3 Comments in pdf

Round 2
Reviewer 1 Report
Thank you, the proposed corrections have been taken into account. I have no further comments.